# Effects of Hydrothermal Processing on *Miscanthus × giganteus* Polysaccharides: A Kinetic Assessment

**DOI:** 10.3390/polym14214732

**Published:** 2022-11-04

**Authors:** Sandra Rivas, Valentín Santos, Juan Carlos Parajó

**Affiliations:** 1Faculty of Science, Chemical Engineering Department, University of Vigo (Campus Ourense), Polytechnical Building, As Lagoas, 32004 Ourense, Spain; 2CINBIO, University of Vigo (Campus Lagoas-Marcosende), 36310 Vigo, Spain

**Keywords:** *Miscanthus*, autohydrolysis, kinetic modeling, polysaccharides, value-added compounds

## Abstract

*Miscanthus × giganteus* samples were characterized for composition and treated with hot compressed water (hydrothermal or autohydrolysis treatments) at temperatures in the range of 190–240 °C. The liquid phases from treatments were analyzed to assess the breakdown of susceptible polysaccharides into a scope of soluble intermediates and reaction products. The experimental concentration profiles determined for the target compounds (monosaccharides, higher saccharides, acetic acid and sugar-decomposition products) were interpreted using a pseudohomogeneous kinetic mechanism involving 27 reactions, which were governed by kinetic coefficients showing an Arrhenius-type temperature dependence. The corresponding activation energies were calculated and compared with data from the literature. The kinetic equations allowed a quantitative assessment of the experimental results, providing key information for process simulation and evaluation.

## 1. Introduction

Lignocellulosic materials (LCMs) are potential renewable feedstocks for the chemical industry, as they provide a sustainable alternative to the dwindling fossil resources and circumvent issues such as the environmental hazards related to global warming, the limitation of supplies, and the volatility of prices.

Among the methods proposed for processing LCMs, the ones based on the biorefinery concept [1] are expected to extract added value from the feedstocks by achieving their selective fractionation into cellulose, hemicelluloses and lignin and by allowing the separate conversion of the latter into commercial products. In this context, LCMs are expected to become an essential resource for the production of energy, chemicals, and materials in the near future [2]. Interestingly, integrated biorefineries (exploiting the diverse components of LCMs) are compatible with the principles of circular economy [3]. In particular, modern and sustainable biorefinery technologies have been claimed to pave the path towards circular economy through effective waste management [4].

Desirably, biorefinery technologies should follow the principles of green chemistry, including the utilization of efficient technologies and minimization of waste generation. LCM biorefineries may include the following sequence of steps: (a) fractionation of lignocellulose into its main polymers (cellulose, hemicelluloses, and lignin); (b) depolymerization of biopolymers; (c) transformation of the monomers into value-added products [5].

Some LCM fractionation methods include a first stage performed in dilute acids or alkaline media [6]. Alternatively, treatments performed with hot compressed water (autohydrolysis or hydrothermal processing) have been employed to cause the partial hydrolysis of hemicelluloses into both soluble saccharides of lower molecular mass and products derived from them, leaving both cellulose and lignin in the solid phase. Interestingly, the solids from hydrothermal treatments show an increased susceptibility to further processing (for example, delignification or enzymatic hydrolysis) [7,8], enabling an integral benefit of the feedstocks. The efficient use of the entire biomass is expected to increase the overall feedstock value and significantly contribute to process cost-effectiveness [9]. Following this idea, the Agenda 2020 Technology Alliance has advocated for the modification of the conventional pulping that is processed by including a hemicellulose extraction stage prior to pulping.

In previous studies, autohydrolysis (in which LCMs and water are the only reagents) has been considered one of the most appropriate choices for hemicellulose fractionation [10,11] due to its environmentally friendly character, high efficiency, low cost, and ability to perform under mild pH conditions (which reduce both equipment and operational costs) [12,13].

During autohydrolysis, hemicelluloses are progressively hydrolyzed into low-molecular polymers, oligomers, monosaccharides, and monosaccharide-derived products [6,14,15,16] through reactions catalyzed by hydronium ions. In this way, a number of target products can be obtained from hemicelluloses, including bioactive oligosaccharides [6,13,17], monosaccharides [18], and furans [19], by tuning the severity of the operational conditions.

According to the above ideas, detailed knowledge of the effects of operational conditions (principally, temperature and reaction time) on the product distribution is valuable, since information on the mechanisms and kinetics of polysaccharide breakdown has been considered essential for process design and optimization [5].

A number of approaches have been employed to assess the conversion of hemicelluloses through autohydrolysis, including:optimization based on the analysis of time–concentration profiles obtained under isothermal conditions [17,20,21,22,23];data interpretation based on parameters measuring the combined effects of temperature and time, such as the P factor [24] or the severity factor [13,15,16,25,26,27,28,29];empirical modeling based on the Surface Response Methodology or fuzzy neural networks [30,31,32,33,34,35,36,37];modeling based on kinetic mechanisms involving multiple reactions (including the breakdown of native polysaccharides into intermediates and the further reactions thereof) [14,15,19,28,38,39,40,41,42,43,44,45,46,47,48,49,50,51,52,53,54,55].

The above studies have been performed for various LCMs, including: (a) woods and woody legumes [16,19,21,22,23,24,27,32,33,34,39,42,44,49,50]; (b) seeds, straws, shells, hulls, and various agricultural byproducts or wastes [13,14,21,25,26,28,37,40,41,43,45,46,47,48,52,53,54,55]; (c) industrial byproducts [30,35,45,51]; (d) herbs and grasses, including dedicated energy crops [15,17,20,21,26,29,36].

Within the latter group, perennial grasses have emerged as potential raw materials for biorefineries with the aim of the sustainable production of renewable fuels and chemicals via thermo-chemical conversion [11], with the advantage that they do not compete with food crops [56]. *Miscanthus* is a non-invasive perennial grass that is able to grow with a remarkable productivity (15–25 tons of dry matter/ha) and with little or no herbicide or nitrogen requirements [20], good tolerance to temperature, and high yield/energy content [57]; thus, it exhibits great potential as a feedstock for conversion into biobased products in integrated biorefineries [2]. In Europe, *Miscanthus* has been identified as a particularly promising raw material for biofuels, as it is capable of economically substituting significantly more than 10% of fossil fuels for transportation [58]. Currently, *Miscanthus* plantations cover about 20,000 ha of the European lands [59].

This work deals with the kinetic modeling of the multiple reactions involved in the fractionation of *Miscanthus × giganteus* (M *×* G) through hydrothermal processing. Based on a mechanism involving series and parallel reactions, the experimental concentration profiles obtained for the most important compounds present in the reaction media (see Table 1 for nomenclature) allowed the calculation of both the preexponential factors and the activation energies for the 27 reactions that described the fate the feedstock polysaccharides. The information obtained in this study enables a deep understanding of fractionation and provides a quantitative interpretation of data, as well as basic information for optimization, evaluation, and scaling-up M × G autohydrolysis.

## 2. Materials and Methods

### 2.1. Raw Material

M × G samples, which were provided by Premier Green Energy (Thurles, Co. Tipperary, Ireland), were ground and sieved to select the fraction of particles with sizes in the range of 0.25–2 mm. The selected lot was air-dried, homogenized, and stored until use.

### 2.2. Autohydrolysis of M × G

M × G samples were mixed with tap water in a 0.6 L stainless steel reactor (Parr Company, Illinois, USA) at a liquid-to-solid ratio (LSR) of 10 kg/kg, and the suspension was heated under stirring (150 rpm) until reaching the target temperature (in the range 190–240 °C) while following the profile shown in Figure 1. The hot compressed water was responsible for the hydrolytic effects on hemicelluloses and cellulose. After reaching the desired temperature, the reactor was cooled immediately (i.e., there was no isothermal reaction period). After cooling, the solid and liquid phases were separated through filtration and assayed for composition and solid yield.

### 2.3. Analytical Methods

Before analytical determinations, the M × G samples were ground in a Wiley mill equipped with a 0.5 mm sieve (Polymix^®^ PX-MFC 90 D, Kinematica AG, Littau, Lucerne, Switzerland). The analyses were as follows: extractives, NREL/TP-510-42619 method; uronic acids, Blumenkrantz and Asboe–Hansen method [60]; structural carbohydrates and lignin in solid samples, NREL/TP-510-42618 protocol (based on a quantitative acid hydrolysis, denoted by QAH). The solid residues from QAH were considered as Klason lignin. The liquid phases from QAH were analyzed with HPLC using a 1200 series instrument (Agilent Technologies, Santa Clara, California, USA) equipped with refraction index (RI) and ultraviolet–diode array (UV-DAD) detectors. Acetic acid, furfural, and hydroxymethylfurfural (HMF) in liquid QAH samples were quantitated using a 300 × 7.8 mm Aminex HPX-87H column (BioRad Life Science Group Hercules, California, USA) operated at 50 °C (mobile phase, 3 mN sulfuric acid; flow, 0.6 mL·min^−1^). Additionally, liquid aliquots from QAH were neutralized with barium carbonate and employed for the determination of xylose, arabinose, glucose, mannose, and galactose using a CARBOSep CHO-682 column (Transgenomic Inc., Omaha, NE 68164, USA) operated at 80 °C (mobile phase, deionized water; flow, 0.4 mL·min^−1^). The composition of the autohydrolysis liquors was determined through HPLC analysis of samples obtained either directly or after a quantitative posthydrolysis performed for 20 min in media containing 4% H_2_SO_4_. Both types of samples were assayed for monosaccharides, furans, and organic acids by using the same HPLC method as that described above to assess the composition of the liquid phases from QAH. The differences in the monosaccharides and acetic acid concentrations between non-posthydrolyzed and posthydrolyzed samples allowed the calculation of the amounts of oligosaccharides and bound acetyl groups. The amounts of non-volatile compounds (NVCs) in the liquid samples from autohydrolysis were measured by oven-drying at 105 °C until reaching a constant weight.

### 2.4. Fundamentals of the Mathematical Modeling

Based on the literature [14,16,19,61], the reactions involving the breakdown of native polysaccharides into soluble oligomers and the further reactions thereof were assumed to be governed by coefficients following an Arrhenius-type temperature dependence:(1)$$k_{i}=k_{0i}\cdot exp\left(\frac{Ea_{i}}{R\cdot T}\right)$$
where *k_i_* is the kinetic coefficient measuring the consumption of the compound *i*, *k_0i_* is the pre-exponential factor, *Ea_i_* is the activation energy, and *T* is the temperature (K).

Based on various hypotheses, a number of kinetic mechanisms were considered. In each case, the differential equations resulting from the respective hypotheses were solved using a 4th-order Runge–Kutta method coupled with the Solver routine built into the Excel software (Microsoft Corporation, Alburquerque, New Mexico, USA). The considered mechanisms were compared on the basis of their accuracy to describe the concentration profiles of the diverse autohydrolysis products.

## 3. Results

### 3.1. Composition of the Raw Material

M × G, a sterile hybrid from *Miscanthus sacchariflorus* and *Miscanthus sinensis*, is a perennial grass that, as a fast-growing crop, is suitable for manufacturing biofuels and chemicals [11] by following the biorefinery concept [62]. Considered as an industrial feedstock, *Miscanthus* presents a number of advantages, including a high carbon dioxide fixation rate, low nitrogen and herbicide requirements, a long lifespan, low susceptibility to pests and diseases, a low moisture content at harvest, and a favorable composition [11,62,63].

As a typical lignocellulosic material, M × G biomass is made up of non-structural components (mainly extractives) and three major polymers (lignin, cellulose, and hemicelluloses, usually termed “structural components”). The chemical nature of these fractions can be described as follows:Extractives are mainly formed by fatty acids, sterols, and other aromatic compounds, for which pharmaceutical applications have been proposed [11].Lignin is an amorphous, tridimensional polymer containing guaiacyl, syringyl, and p-hydroxyphenyl structural units derived from coniferyl, sinapyl, and p-coumaryl alcohols, respectively.Cellulose is a linear polymer made up of β-D-glucopyranose units.

Hemicelluloses corresponded mostly to heteroxylan, a linear β-1,4-linked backbone of xylosyl residues substituted by 4-*O*-methylglucuronic acid or arabinose and acetyl groups [17,29,63]. As with other herbaceous materials [64], M × G may contain minor amounts of galactan and mannan.

Wide ranges of variation have been reported for the relative abundance of the major components of *Miscanthus*. The authors of [11,65] indicated the following ranges of variation: cellulose, 40–60 wt%; hemicelluloses, 20–40%; lignin, 10–30%. The compositional results determined in this work are listed in Table 2, together with data reported in related studies.

The results in Table 2 show that the sample employed in this study contained typical amounts of cellulose and lignin, with a comparatively low extractive content, as well as hemicelluloses dominated by xylan. As an additional result that was necessary for kinetic modeling (not included in Table 2), the content of uronyl substituents was 3.03%. As a whole, the hemicellulose fraction (measured by the joint contributions of xylosyl, galactosyl, arabinosyl, mannosyl, acetyl, and uronyl components) accounted for 30.6% of the oven-dried raw material. From these results, it could be calculated that the molar ratios of the xylosyl:arabinosl:acetyl:uronyl units were 1:0.13:0.48:0.10.

### 3.2. Experimental Results Obtained in Hydrothermal Treatments

The hydrothermal processing of xylan-containing LCMs is known to cause a number of structural, physical, and chemical modifications, including a decrease in cellulose crystallinity and accessibility, depolymerization of polysaccharides, selective solubilization of the hemicellulose fraction through partial hydrolysis, generation of acetic acid from acetyl groups [26], removal of water-soluble extractives, release of sterified phenolic acids, and decrease in aliphatic hydroxyl groups and β-*O*-4 structures in the residual lignin [24].

Among these effects, the reactions involving cellulose and hemicelluloses have received special attention, as they enable the design of selective processes aimed at an integral benefit of LCMs.

According to the information in Table 1, the solid and liquid phases were characterized in terms of the following components:Solid substrates: glucosyl units in cellulose (Gl_n_), xylosyl units in xylan (X_n_), galactosyl units in hemicelluloses (Ga_n_); arabinosyl units in hemicelluloses (Ar_n_); acetyl groups attached to residual hemicelluloses (Ac_n_).Liquid phases: glucosyl units in soluble low-molecular-weight polymers or oligosaccharides (GlPOS); xylosyl units in low-molecular-weight polymers or oligosaccharides (XyPOS), galactosyl units in low-molecular-weight polymers or oligosaccharides (GaPOS), arabinosyl units in low-molecular-weight polymers or oligosaccharides (ArPOS), acetyl groups attached to soluble low-molecular-weight polymers or oligosaccharides (AcPOS), uronyl groups in soluble compounds (UA), glucose (Glu), xylose (Xyl), galactose (Gal), arabinose (Ara), furfural (F), 5-hydroxymethyl furfural (HMF), and acetic acid (AcH).

For calculation purposes, the anhydrosugar units in polymers or oligosaccharides were expressed as hexose or pentose equivalents. In the same way, acetyl and uronyl groups were expressed as acetic acid and uronic acid equivalents, respectively. Mannosyl units and their derived products were not considered in this study due to their low concentrations.

Table 3 lists data regarding the composition of the solid and liquid phases with respect to the temperature and reaction time. A number of general conclusions can be drawn from the set of experimental data, namely, the scarce solubilization of cellulose at temperatures below 200 °C, the fast solubilization of xylan and xylan substituents (arabinosyl and acetyl groups, which were almost completely removed from the solid phase at 230 °C), and the comparatively higher stability of uronyl substituents. Regarding the components present in the liquid phase, all of the polymeric or oligomeric components (GlPOS, XyPOS, GaPOS, ArPOS, AcPOS, and UA) behaved as reaction intermediates, reaching maximum concentrations at specific temperatures within the considered range. Diverse reaction patterns were observed for monosaccharides, among which Ara and Gal showed clearly defined concentration maxima, confirming their comparatively high susceptibility toward decomposition. Oppositely, the concentrations of AcH, F, and HMF increased steadily with temperature.

### 3.3. Fundamentals of Kinetic Modeling

As explained in the introduction, studies dealing with the interpretation of LCM autohydrolysis include both empirical assessments and kinetic studies based on pseudohomogeneous kinetics. This work follows the latter approach due to its suitability for optimization and design calculations.

Earlier studies on the depolymerization of xylan upon autohydrolysis [38,39,40,41,42,43,55] were based on the following hypotheses:Xylan can be made up of one or two fractions with different susceptibilities to hydrolysis.Susceptible xylan is progressively broken down into soluble oligomers of decreasing molecular weight (with low-molecular-weight fractions derived from high-molecular-weight fractions), whereas the non-susceptible fraction may or may not contribute to the generation of oligosaccharides.Xylose is first generated from low-molecular-weight oligosaccharides and then dehydrated into furfural.Furfural can be produced directly from oligosaccharides.Under harsh conditions, furfural can be converted into degradation products.

Other studies considered the existence of up to three xylan fractions [53,54] that led to one or more types of oligosaccharides and/or paid attention to the fate of other hemicellulose components, such as arabinan, acetyl groups, and/or uronyl substituents [14,15,51,53,54]. Other studies proposed alternative mechanisms involving the direct formation of xylose from susceptible xylan [50], the production of furfural from arabinose [47], the formation of both furfural and degradation products from xylose [49], the hydrolysis of cellulose into oligosaccharides, and the further reactions of these latter compounds [46,52], as well as the dissolution of solid material [44]. More recently, kinetic assessments based on the above ideas were reported for a number of lignocellulosic substrates, including *Annona cherimola* seeds [28], birch wood [19], vine shoots [48], wheat straw, corn stover, and sugarcane bagasse [45].

In this work, a number of models were considered for the interpretation of the data, assuming one or two fractions for cellulose, xylan, and uronyl groups attached to xylan, reaction orders of 1 or 2, and the presence or absence of reactions leading to the formation of degradations products from Glu, Gal, Xyl, Ara, UA, F, and HMF. The simplest mechanism providing a satisfactory interpretation of the data was based on the following assumptions:All of the reactions considered were of the first order, with kinetic coefficients showing an Arrhenius-type dependence on temperature,The participation of cellulose hydrolysis reactions was significant under the conditions tested. Cellulose (glucan, Gl_n_) is made up of two fractions (fast-reacting glucan, denoted as Gl_nf_, and slow-reacting glucan, denoted as Gl_ns_), with relative proportions measured by the “susceptible fraction” α_Gln_, which is defined as the g of Gl_nf_ per 1 g of Gl_n_.GlPOSs are generated from both cellulose fractions and converted into Glu, which reacts to yield HMF or degradation products (denoted DP_Glu_).Ga_n_ is converted into GaPOS through a single reaction. GaPOSs are converted into Gal, which reacts to yield HMF or degradation products (denoted as DP_Gal_). For the sake of simplicity, the concentrations of HMF were measured in terms of hexose equivalents.Xylan is made up of two fractions (fast-reacting xylan, denoted as X_nf_, and slow-reacting xylan, denoted as X_ns_), with relative proportions measured by the “susceptible fraction” α_Xn_, which is defined as the g of X_nf_ per g of X_n_.XyPOSs are generated from both xylan fractions and converted into Xyl, which reacts to yield F or degradation products (denoted as DP_Xyl_).Ar_n_ is converted into ArPOS through a single reaction. ArPOSs are converted into Ara, which reacts to yield F or degradation products (denoted DP_Ara_). For the sake of simplicity, the concentrations of F are measured in terms of equivalent pentoses.Upon reaction, the acetyl groups in hemicelluloses (Ac_n_) appear first while attached to oligosaccharides (AcPOS) and are further hydrolyzed into acetic acid (AcH)The uronyl groups in hemicelluloses (UA_n_) are present in the soluble hydrolysis products (UA), which react to give decomposition products (DP_UA_)

Figure 2 summarizes the resulting set of individual reactions, as well as the nomenclature employed for the respective kinetic coefficients.

### 3.4. Kinetic Modeling of Reactions Involving Hexoses

The hypotheses explained above for the solubilization of Gl_n_ and Ga_n_ lead to the following set of equations:(2)$$Gl_{n}=Gl_{nf}+Gl_{ns}$$
(3)$$\alpha_{Gln}={\frac{Gl_{nf}}{Gl_{n}}}_{}$$
(4)$$\frac{d\left[Gl_{nf}\right]}{dt}=-k_{Glnf}\cdot \left[Gl_{nf}\right]$$
(5)$$\frac{d\left[Gl_{ns}\right]}{dt}=-k_{Glns}\cdot \left[Gl_{ns}\right]$$
(6)$$\frac{d\left[GlPOS\right]}{dt}=k_{Glnf}\cdot \left[Gl_{nf}\right]+k_{Glns}\cdot \left[Gl_{ns}\right]-k_{GlPOS}\cdot \left[GlPOS\right]$$
(7)$$\frac{d\left[Glu\right]}{dt}=k_{GlPOS}\cdot \left[GlPOS\right]-k_{Glu}\cdot \left[Glu\right]-k_{2Glu}\cdot \left[Glu\right]$$
(8)$$\frac{d\left[Ga_{n}\right]}{dt}=-k_{Gan}\cdot \left[Ga_{n}\right]$$
(9)$$\frac{d\left[GaPOS\right]}{dt}=k_{Gan}\cdot \left[Ga_{n}\right]-k_{GaPOS}\cdot \left[GaPOS\right]$$
(10)$$\frac{d\left[Gal\right]}{dt}=k_{GaPOS}\cdot \left[GaPOS\right]-k_{Gal}\cdot \left[Gal\right]-k_{2Gal}\cdot \left[Gal\right]$$
(11)$$\frac{d\left[HMF\right]}{dt}=k_{Glu}\cdot \left[Glu\right]+k_{Gal}\cdot \left[Gal\right]$$

The preexponential factors, the activation energies, and the susceptible glucan fraction obtained by fitting the experimental data to the above equations are listed in Table 4.

The suitability of the proposed mechanism for the interpretation of the data can be assessed from the results in Figure 3, which shows the experimental and calculated concentration profiles for the polysaccharides made up of hexoses (Gl_n_ and Ga_n_) and the products derived from them.

Scarce information has been reported on the fate of cellulose during autohydrolysis treatments, which are usually focused on the selective solubilization of hemicelluloses. Sidiras et al. [52] determined the optimal conditions for glucose production (240 °C, 82 min) from wheat straw cellulose and proposed a kinetic mechanism based on the separate production of a single type of soluble oligomers from crystalline cellulose and amorphous cellulose. The mechanism also considered reactions leading to the formation of Glu and HMF. The result reported for the activation energy of cellulose hydrolysis (116.4 kJ/mol) was identical to the result achieved in this work for the solubilization of Gl_nf_, whereas the activation energies of the reactions involving the formation of glucose and its conversion into HMF (125 and 164 kJ/mol) were higher than the corresponding values listed in Table 4. In general, broad ranges of variation have been reported for the conversion of cellulose into soluble oligosaccharides. For example, the activation energy for the conversion of the susceptible fraction of cellulose in rye straw (accounting for 93.8% of the total cellulose) into soluble oligosaccharides was 49.9 kJ/mol [47], whereas a higher value (Ea = 325.7 kJ/mol) was reported for the conversion of the whole cellulose in pine wood [16].

The differences observed were ascribed to the diverse properties of the lignocellulosic substrates, since pine wood is a strongly lignified material (a factor affecting the accessibility of cellulose), and its cellulose fraction presents a comparatively high crystallinity. The same authors reported on the modeling of galactosyl group hydrolysis, and they used *Pinus pinaster* wood as a raw material. Ga_n_ was assumed to be hydrolyzed by two parallel reactions, one leading to GaOS (with E_a_/R = 1.70 × 10^4^ K) and the other yielding Gal.

### 3.5. Kinetic Modeling of Reactions Involving Pentoses

The experimental data in Figure 4 show a sharp decrease in the amount of xylosyl groups in solids with the temperature, particularly in the experiments performed at 195–215 °C. According to the scheme in Figure 2, both xylan fractions were hydrolyzed into XyPOSs, which reached a maximum concentration (10.7 g/L) at 207 °C, and they were split into xylose units, which reached comparatively low concentrations (maximum value: 2.17 g/L at 215 °C), owing to their fast dehydration into furfural.

High temperatures led to the conversion of XyPOSs into Xyl and to the formation of F from pentoses. Xyl (potential concentration: 21.9 g/L) was the major sugar present in autohydrolysis liquor, and it reached its maximum concentration (2.17 g/L) at 215 °C (Figure 4).

A number of kinetic mechanisms have been proposed to assess the autohydrolysis of xylan-containing materials. The hypotheses employed depended on both the type of feedstock and the operational conditions. Studies dealing with *Pinus pinaster* wood [16] and sugarcane bagasse [61] assumed that all of the xylan in the feedstocks presented the same susceptibility to hydrolysis, behaving as a single fraction that was decomposed into oligosaccharides through reactions with activation energies of 208.6 and 142.1 kJ/mol, respectively. Most cases assumed that the feedstock contained two xylan fractions, one of which was easily hydrolyzable, and the other may or may not undergo hydrolysis under the conditions assayed. Studies in which it was assumed that a part of xylan was unreactive employed barley husks [14], *Arundo donax* [15], birch [19], *Eucalyptus globulus* [39], corncobs [40], *Acacia dealbata* [44], rye straw [47], vine shoots [48], sugar maple wood [50], a brewery’s spent grains [51], and sugarcane bagasse [74].

According to the mechanism proposed in this study, the reactions describing the conversion of *X_n_* and *Ar_n_* in autohydrolysis media are as follows:(12)$$X_{n}=X_{ns}+X_{ns}$$
(13)$$\propto_{Xn}=[\frac{X_{nf}}{X_{n}}{]}_{RM}$$
(14)$$\frac{d\left[X_{nf}\right]}{dt}=-k_{Xnf}\cdot \left[X_{nf}\right]$$
(15)$$\frac{d\left[X_{ns}\right]}{dt}=-k_{Xns}\cdot \left[X_{ns}\right]$$
(16)$$\frac{d\left[XPOS\right]}{dt}=k_{Xnf}\cdot \left[X_{nf}\right]+k_{Xns}\cdot \left[X_{ns}\right]-k_{XyPOS}\cdot \left[XyPOS\right]$$
(17)$$\frac{d\left[Xyl\right]}{dt}=k_{XyPOS}\cdot \left[XyPOS\right]-k_{Xyl}\cdot \left[Xyl\right]-k_{2Xyl}\cdot \left[Xyl\right]$$

Table 5 includes the results calculated for α*_Xn_* and for the activation energies and preexponential factors involved in the above equations.

In the literature [14,15,19,39,40,44,47,48,50,51,74], the mass fraction corresponding to fast-reacting xylan varied from 0.708 (for experiments performed with breweries’ spent grains at low temperatures) to 0.917 (employing sugarcane bagasse). In turn, the activation energy of the kinetic coefficient governing the depolymerization of the susceptible xylan varied within the range of 65.6–225.2 kJ/mol, with the lowest values corresponding to strongly acetylated hardwoods. In comparison, values from 125 to 251.7 kJ/mol were reported for the activation energy of the slow-reacting fraction [38,41,49,52,53,55].

Concerning arabinosyl groups and the products derived from them, small amounts of arabinose were found in the reaction media due to its low potential concentration (2.80 g/L). The experimental concentration profiles determined for the products derived from arabinosyl groups are shown in Figure 5. Ar_n_ was depleted at 215 °C, whereas the maximum concentrations of ArPOS and Ara (0.75 and 0.72 g/L, respectively) were achieved at 195 and 202 °C, respectively. These data confirm the higher lability reported for arabinosyl groups in comparison with that of xylosyl groups [5,26]. F was obtained through the dehydration of both Ara and Xyl, reaching a maximum concentration of 2.74 g/L (corresponding to a 17.3% molar yield).

On the basis of the proposed mechanism, the following set of equations describing the conversion of Ar_n_ and the derived product can be derived:(18)$$\frac{d\left[Ar_{n}\right]}{dt}=-k_{Arn}\cdot \left[Ar_{n}\right]$$
(19)$$\frac{d\left[ArPOS\right]}{dt}=k_{Arn}\cdot \left[Ar_{n}\right]-k_{ArPOS}\cdot \left[ArPOS\right]$$
(20)$$\frac{d\left[Ara\right]}{dt}=k_{ArPOs}\cdot \left[ArPOS\right]-k_{Ara}\cdot \left[Ara\right]-k_{2Ara}\cdot \left[Ara\right]$$
(21)$$\frac{d\left[F\right]}{dt}=k_{Xyl}\cdot \left[Xyl\right]+k_{Ara}\cdot \left[Ara\right]$$

The solution of the above equations led to the set of parameters listed in Table 6, whereas the agreement between the experimental and calculated concentration profiles can be assessed in Figure 5.

In the literature, the effects of autohydrolysis on the arabinosyl groups present in various lignocellulosic materials were reported, and their hydrolysis into soluble products that make parts of sugar oligomers was assumed. This hypothesis was followed in studies dealing with *Arundo donax* [15], spent grains from breweries [51], barley husks [14], *Pinus pinaster* [16], vine shoots [48], birch wood [19], and *Acacia dealbata* [44]. In these studies, the activation energy of the reaction describing the solubilization of acetyl groups was reported to fall in the range of 95.7–216 kJ/mol.

### 3.6. Kinetic Modeling of Reactions Involving Acetyl and Uronyl Groups

In this study, the acetyl groups present in hemicelluloses were assumed to be present in acetylated oligomers and transformed into acetic acid upon further hydrolysis. According to these ideas, the following equations can be derived:(22)$$\frac{d\left[Ac_{n}\right]}{dt}=-k_{Acn}\cdot \left[Ac_{n}\right]$$
(23)$$\frac{d\left[AcPOS\right]}{dt}=k_{Acn}\cdot \left[Ac_{n}\right]-k_{AcPOS}\cdot \left[AcPOS\right]$$
(24)$$\frac{d\left[AcH\right]}{dt}=k_{AcPOS}\cdot \left[AcPOS\right]-k_{AcH}\cdot \left[AcH\right]$$

Table 7 includes the values of the fitting parameters obtained from the above equations, whereas Figure 6 allows the comparison between the experimental and calculated concentration profiles. AcPOSs behaved as reaction intermediates, reaching a maximum concentration of 1.19 g/L. In turn, no acetic acid consumption by side reactions was observed. Acetic acid reached a maximum concentration of 3.90 g/L under the harshest conditions assayed. The literature in which reactions involving acetyl groups in autohydrolysis media were reported followed two different approaches: Either just a fraction of the acetyl group was reactive under the considered conditions, or all of the acetyl groups were susceptible to the reaction. The first approach was followed in works using *Arundo donax* [15], hardwoods and corncobs [26], rye straw [47], and vine shoots [48]; in those studies, the susceptible fraction accounted for 0.825–0.941 g/g of acetyl groups, and the corresponding activation energies were within the range of 103–274 kJ/mol. The one-fraction approach was followed in studies dealing with barley husks [14], *Pinus pinaster* [16], wheat straw [52], and *Acacia dealbata* [44], which presented activation energies in the range of 58–119 kJ/mol.

Concerning the uronyl groups attached to soluble compounds, the maximum experimental concentration reached 0.49 g/L. The experimental concentration profile showed a well-defined maximum, confirming the importance of decomposition reactions.

The hypotheses listed above led to the following equations for the solubilization and degradation of uronyl moieties:(25)$$\frac{d\left[UA_{ns}\right]}{dt}=-k_{UAns}\cdot \left[UA_{ns}\right]$$
(26)$$\frac{d\left[UA\right]}{dt}=k_{UAns}\cdot \left[UA_{ns}\right]-k_{UA}\cdot \left[UA\right]$$
(27)$$DP_{UA}=UAn_{RM}-UA$$

Table 8 lists the set of regression parameters resulting from data fitting, whereas Figure 7 confirms the ability of the mathematical model to give a quantitative interpretation of the data.

In comparative terms, scarce attention has been devoted in the literature to the elucidation of the fate of uronyl groups during autohydrolysis. The available studies [15,47] that were performed with *Arundo donax* and rye straw assumed that some uronyl groups did not react under the conditions tested, whereas the susceptible fraction (accounting for 0.893–0.939 g/g of uronyl groups) was solubilized through a reaction showing activation energies of 141 and 158 kJ/mol, respectively.

## 4. Conclusions

This study presents a sound assessment of the kinetics of the chemical modifications undergone by M × G biomass during processing with non-isothermal autohydrolysis. Various experiments were performed under selected operational conditions (covering the range of practical interest) to establish their effects on the concentration profiles of the target products (soluble oligomers derived from cellulose and hemicellulosic polymers, monosaccharides, acetic acid, uronyl groups, furans, and degradation products). The experimental data were fitted to equations derived from a kinetic model involving 27 reactions, which provided a satisfactory interpretation of the experimental data and provided key information for process design and evaluation.

## Figures and Tables

**Figure 1 polymers-14-04732-f001:**
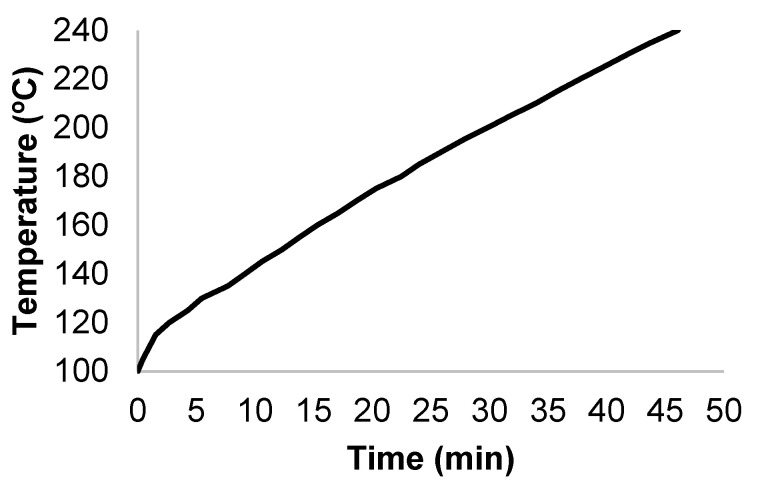
Temperature profile that was followed in autohydrolysis experiments.

**Figure 2 polymers-14-04732-f002:**
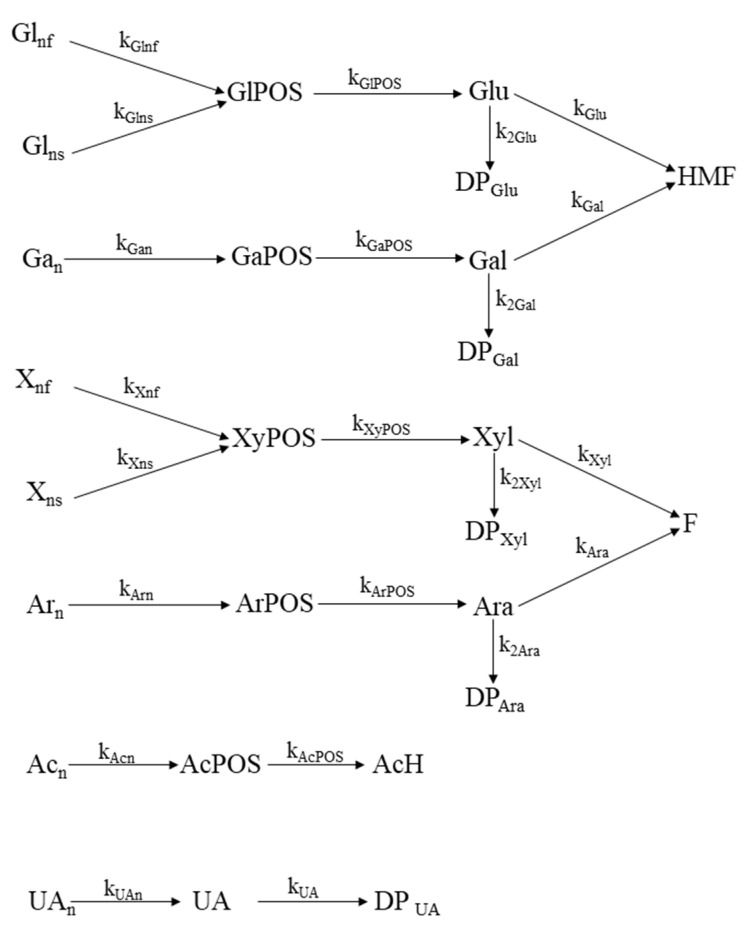
Mechanism proposed for the kinetic modeling of MxG autohydrolysis.

**Figure 3 polymers-14-04732-f003:**
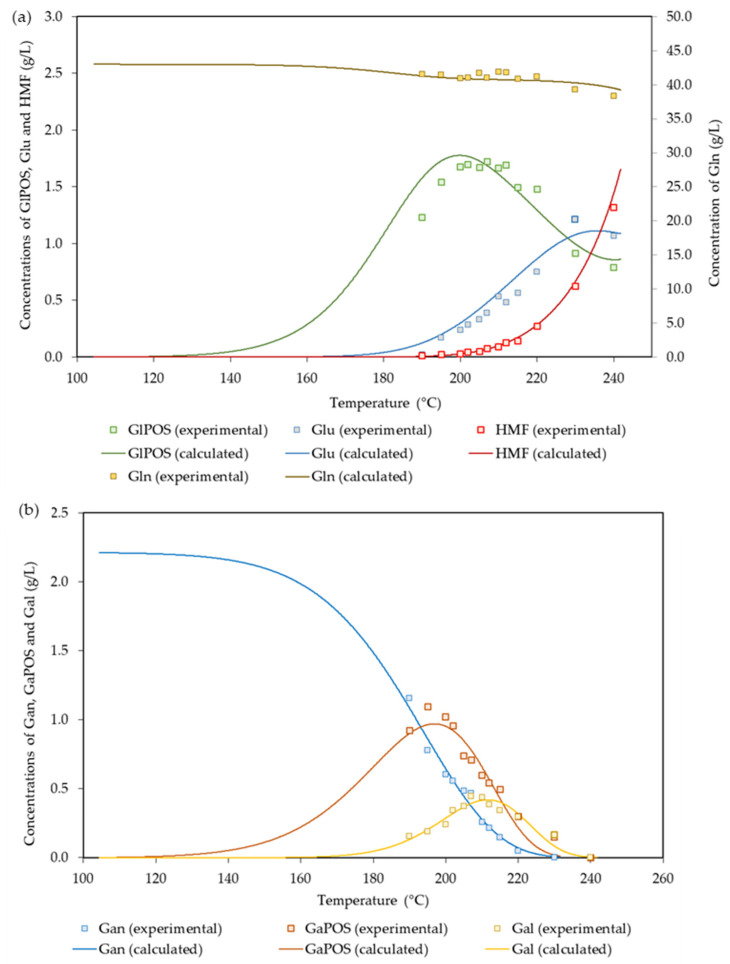
Experimental and calculated values determined for: (**a**) Gl_n_ and the derived products; (**b**) Ga_n_ and the derived products.

**Figure 4 polymers-14-04732-f004:**
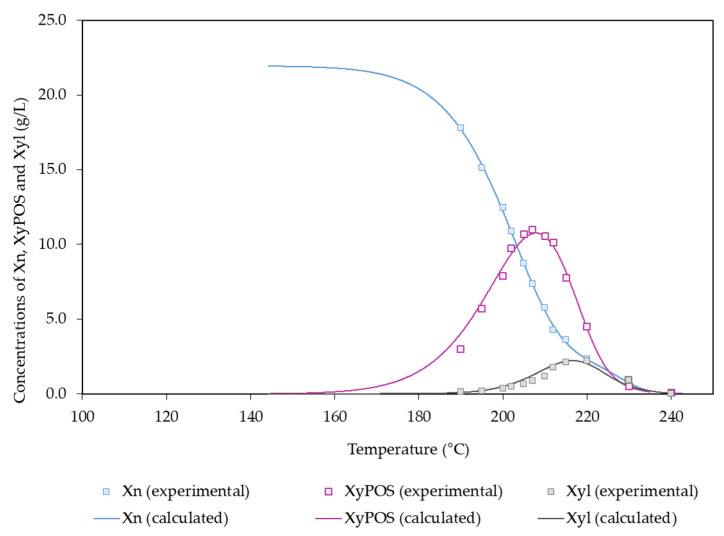
Experimental and calculated values determined for X_n_ and the derived products.

**Figure 5 polymers-14-04732-f005:**
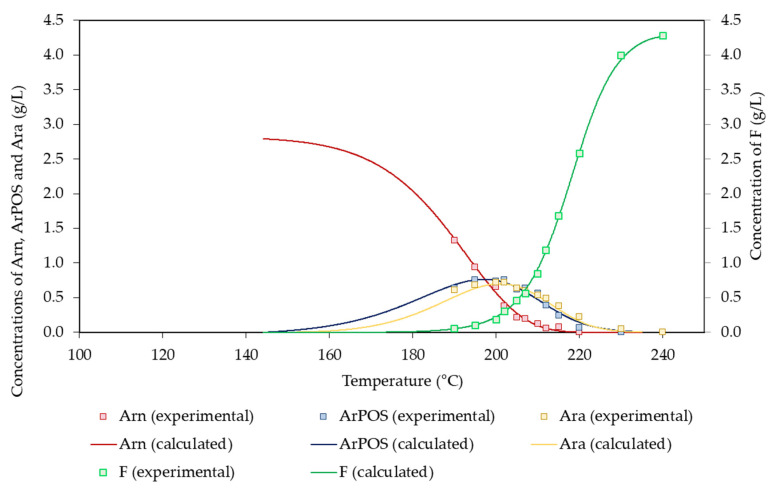
Experimental and calculated values determined for arabinosyl groups and the derived products.

**Figure 6 polymers-14-04732-f006:**
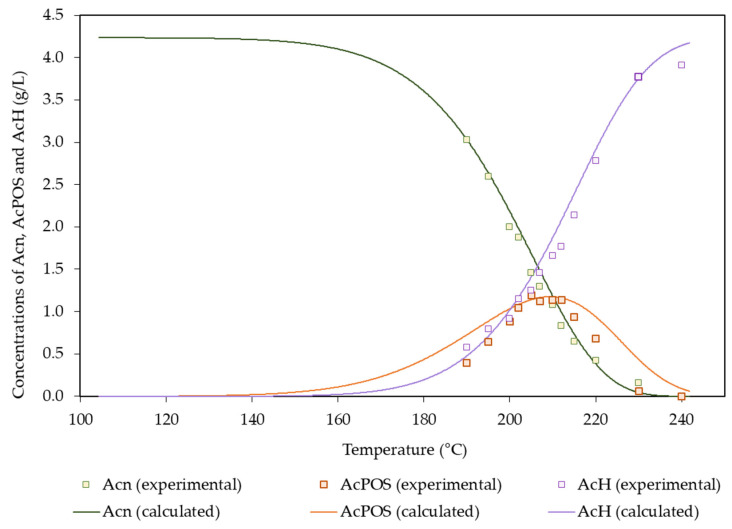
Experimental and calculated concentration profiles determined for acetyl groups and acetic acid.

**Figure 7 polymers-14-04732-f007:**
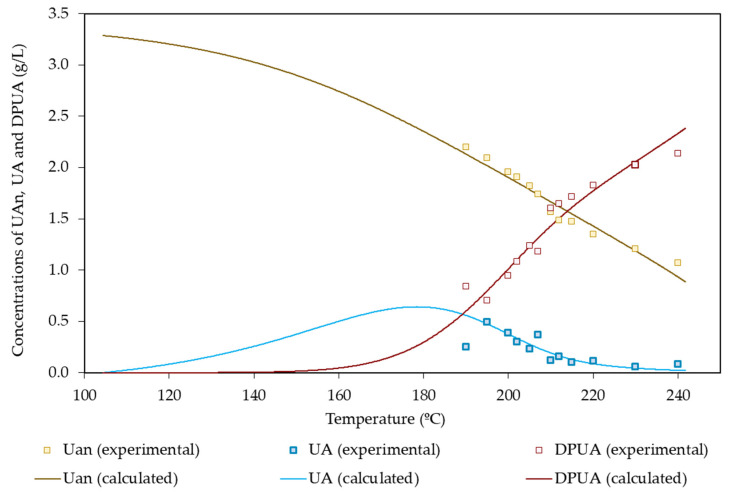
Experimental and calculated values of variables measuring the hydrolysis of uronic acids.

**Table 1 polymers-14-04732-t001:** Nomenclature employed in the kinetic modeling of M × G autohydrolysis.

COMPONENT	NOMENCLATURE	COMPONENT	NOMENCLATURE
Glucosyl units in polysaccharides (glucan)	Gl_n_	Galactosyl units oligosaccharides	GaPOS
Fast-reacting glucan	Gl_nf_	Acetyl groups in oligomers	AcPOS
Slow-reacting glucan	Gl_ns_	Uronyl substituents	UA
Fast-reacting glucan fraction	α_Gln_	Glucose	Glu
Xylosyl units in polysaccharides (xylan)	X_n_	Xylose	Xyl
Fast-reacting xylan	X_nf_	Arabinose	Ara
Slow-reacting xylan	X_ns_	Galactose	Gal
Fast-reacting xylan fraction	α_Xn_	Acetic acid	AcH
Arabinosyl units in polymers	Ar_n_	Furfural	F
Galactosyl units in polymers	Ga_n_	5-Hydroxymethylfurfural	HMF
Acetyl groups in polymers	Ac_n_	Degradation products from glucose	DP_Glu_
Uronyl units in polymers	UA_n_	Degradation products from xylose	DP_Xyl_
Glucosyl units in oligosaccharides	GlPOS	Degradation products from arabinose	DP_Ara_
Xylosyl units in oligosaccharides	XyPOS	Degradation products from galactose	DP_Gal_
Arabinosyl units in oligosaccharides	ArPOS	Degradation products from uronic acids	DP_UA_

**Table 2 polymers-14-04732-t002:** Compositional results reported for *Miscanthus* and the experimental results achieved in this study. Data are expressed as wt% on an oven-dried basis.

Species	Xylan	Arabinan	Galactan	Mannan	Acetyl Groups	Cellulose	Klason Lignin	Ash	Extractives	Reference
*M. sinensis*	22.4 ^(a)^	-	-	-	-	40.2 ^(b)^	24.4	-	-	[66]
M. sinensis	30.3 ^(c)^	-	-	-	-	42.2	19.9	0.7	9.1	[67]
*M. sinensis*	30.56–33.98	-	-	-	-	43.06–52.20	9.23–10.32	2.22–3.47	-	[11]
*M. sacchariflorus*	27.41–28.11 ^(d)^	-	-	-	-	49.06–50.18	12.10–12.13	2.16–2.29	-	[11]
*Miscanthus* sp.	19.0	1.8	0.4	-	-	39.5	24.1	2.0	4.2	[68]
*Miscanthus* sp.	20.75	1.45	-	-	0.11	41.54	20.27	4.24	5.57	[65]
*M × G*	17.1	1.0	0.2	-	-	47.6	22.6	-	11.5	[9]
*M × G*	22.5	2.3	0.4	0.2	-	45.0	26.0	2.8	1.0	[20]
*M × G*	24.83–25.76	-	-	-	-	50.34–52.13	12.02–12.58	2.67–2.74	-	[11]
*M × G*	14.9	1.1	0.3	0	-	38.0	20.8	0.8	4.0	[29]
*M × G*	19.8	-	-	-	-	43.0	25.7	-	-	[69]
*M × G*	19.5	-	0	0	-	35.9	18.5	1.8	11.3	[17]
*M × G*	16.3	2.4	0.46	-	3.10	36.5	13.9	-	-	[70]
*M × G*	20.25	3.98	-	0	2.24	33.96	-	5.50	2.33	[71]
*M × G*	24.3	1.1	0.6	0	-	47.4	25.7	-		[72]
*M × G*	18.8	2.4	0.3	0.1	3.0	43.8	20.8	2.9	6.6	[73]
*M × G*	19.85	1.73	-	-	2.72	42.2	20.57	3.05	6.61	[63]
*M × G*	19.3	2.46	1.99	0.90	2.96	39.0	24.5	3.11	2.75	This work

^(a)^ Measured as total pentoses. ^(b)^ Measured as total hexoses. ^(c)^ Measured by the difference between the contents of holocellulose and alpha-cellulose. ^(d)^ Measured as total hemicelluloses.

**Table 3 polymers-14-04732-t003:** Operational conditions considered in this study, compositions of the treated solids, and concentrations of the liquid phases (measured in g/L).

Operational Conditions	Temp. (°C)	190	195	200	202	205	207	210	212	215	220	230	240
	Time (min)	26.0	27.5	29.5	30.5	31.5	32.5	33.5	34.5	35.5	38.0	42.0	47.0
**Comp.treated solids ± s.d., g/100 g**	**Gl_n_**	43.29 ± 0.20	45.2 ± 0.30	48.27 ± 0.74	47.74 ± 1.27	50.99 ± 2.04	51.40 ± 0.43	54.38 ± 0.99	54.95 ± 0.42	53.76 ± 0.48	56.13 ± 0.02	58.78 ± 0.31	54.50 ± 0.20
	**Ga_n_**	1.26 ± 0.01	0.89 ± 0.05	0.74 ± 0.01	0.68 ± 0.05	0.62 ± 0.03	0.61 ± 0.01	0.35 ± 0.00	0.30 ± 0.00	0.20 ± 0.00	0.00 ± 0.00	0.00 ± 0.00	0.00 ± 0.00
	**X_n_**	19.00 ± 0.01	16.9 ± 0.38	15.02 ± 0.63	12.95 ± 0.08	10.89 ± 0.14	9.42 ± 0.19	7.66 ± 0.09	5.73 ± 0.04	4.85 ± 0.20	3.22 ± 0.17	1.06 ± 0.09	0.00 ± 0.00
	**Ar_n_**	1.41 ± 0.04	1.05 ± 0.02	0.79 ± 0.02	0.45 ± 0.03	0.27 ± 0.01	0.25 ± 0.03	0.17 ± 0.00	0.08 ± 0.01	0.11 ± 0.03	0.00 ± 0.00	0.00 ± 0.00	0.00 ± 0.00
	**Ac_n_**	2.57 ± 0.08	2.30 ± 0.01	1.92 ± 0.01	1.77 ± 0.06	1.44 ± 0.06	1.32 ± 0.05	1.17 ± 0.03	0.89 ± 0.01	0.69 ± 0.02	0.47 ± 0.03	0.18 ± 0.01	0.00 ± 0.00
	**Ua_n_**	2.45 ± 0.05	2.45 ± 0.07	2.47 ± 0.10	2.38 ± 0.10	2.38 ± 0.14	2.33 ± 0.20	2.18 ± 0.04	2.09 ± 0.06	2.08 ± 0.12	2.04 ± 0.18	1.87 ± 0.09	1.69 ± 0.02
	**Klason lignin**	25.75 ± 0.43	27.57 ± 0.30	28.08 ± 0.46	28.58 ± 0.58	27.80 ± 0.16	30.06 ± 0.07	30.27 ± 0.13	31.83 ± 0.20	32.43 ± 0.09	34.66 ± 0.08	37.74 ± 0.11	39.35 ± 0.07
	**Other**	4.32	3.61	2.69	5.43	5.58	4.60	3.85	4.12	5.87	3.47	4.36	4.40
**Conc. Compounds in liquid phase ± s.d., g/L**	**GlPOS**	1.23 ± 0.10	1.54 ± 0.20	1.68 ± 0.07	1.70 ± 0.08	1.67 ± 0.10	1.72 ± 0.10	1.67 ± 0.05	1.69 ± 0.10	1.49 ± 0.09	1.48 ± 0.07	0.92 ± 0.01	0.78 ± 0.02
	**GaPOS**	0.92 ± 0.06	1.09 ± 0.04	1.02 ± 0.03	0.95 ± 0.02	0.74 ± 0.04	0.71 ± 0.05	0.60 ± 0.01	0.54 ± 0.02	0.49 ± 0.02	0.30 ± 0.02	0.15 ± 0.00	0.00 ± 0.00
	**XyPOS**	2.98 ± 0.09	5.71 ± 0.24	7.90 ± 0.13	9.73 ± 0.20	10.66 ± 0.13	10.96 ± 0.11	10.60 ± 0.15	10.13 ± 0.16	7.75 ± 0.12	4.49 ± 0.10	0.51 ± 0.01	0.07 ± 0.01
	**ArPOS**	0.63 ± 0.02	0.75 ± 0.03	0.73 ± 0.04	0.75 ± 0.03	0.62 ± 0.03	0.63 ± 0.02	0.56 ± 0.06	0.38 ± 0.04	0.24 ± 0.01	0.07 ± 0.01	0.00 ± 0.00	0.00 ± 0.00
	**AcPOS**	0.39 ± 0.01	0.64 ± 0.01	0.88 ± 0.06	1.04 ± 0.04	1.18 ± 0.04	1.12 ± 0.07	1.14 ± 0.09	1.14 ± 0.10	0.93 ± 0.01	0.68 ± 0.00	0.06 ± 0.00	0.00 ± 0.00
	**Glu**	0.02 ± 0.00	0.17 ± 0.00	0.24 ± 0.01	0.28 ± 0.01	0.33 ± 0.01	0.38 ± 0.03	0.53 ± 0.02	0.48 ± 0.01	0.56 ± 0.03	0.75 ± 0.01	1.21 ± 0.03	1.06 ± 0.00
	**Gal**	0.15 ± 0.01	0.19 ± 0.00	0.23 ± 0.02	0.34 ± 0.03	0.37 ± 0.01	0.44 ± 0.01	0.43 ± 0.01	0.39 ± 0.02	0.34 ± 0.01	0.30 ± 0.00	0.16 ± 0.01	0.00 ± 0.00
	**Xyl**	0.15 ± 0.00	0.16 ± 0.00	0.32 ± 0.01	0.46 ± 0.04	0.64 ± 0.02	0.84 ± 0.06	1.18 ± 0.07	1.76 ± 0.22	2.11 ± 0.03	2.20 ± 0.05	0.9 ± 0.07	0.00 ± 0.00
	**Ara**	0.61 ± 0.03	0.69 ± 0.05	0.71 ± 0.02	0.72 ± 0.00	0.63 ± 0.02	0.60 ± 0.02	0.54 ± 0.04	0.49 ± 0.03	0.38 ± 0.01	0.22 ± 0.01	0.05 ± 0.00	0.00 ± 0.00
	**AcH**	0.58 ± 0.03	0.79 ± 0.02	0.91 ± 0.03	1.15 ± 0.05	1.25 ± 0.08	1.45 ± 0.07	1.60 ± 0.08	1.77 ± 0.08	2.14 ± 0.12	2.78 ± 0.06	3.77 ± 0.09	3.90 ± 0.06
	**UA**	0.25 ± 0.01	0.49 ± 0.01	0.39 ± 0.03	0.30 ± 0.00	0.23 ± 0.00	0.37 ± 0.03	0.12 ± 0.00	0.16 ± 0.01	0.10 ± 0.00	0.11 ± 0.00	0.06 ± 0.00	0.08 ± 0.01
	**HMF**	0.00 ± 0.00	0.01 ± 0.00	0.02 ± 0.00	0.03 ± 0.00	0.03 ± 0.00	0.05 ± 0.01	0.06 ± 0.00	0.09 ± 0.01	0.10 ± 0.00	0.19 ± 0.01	0.44 ± 0.02	0.92 ± 0.04
	**F**	0.03 ± 0.00	0.07 ± 0.00	0.12 ± 0.01	0.19 ± 0.01	0.29 ± 0.01	0.35 ± 0.02	0.54 ± 0.02	0.76 ± 0.03	1.07 ± 0.04	1.65 ± 0.04	2.56 ± 0.08	2.74 ± 0.07

**Table 4 polymers-14-04732-t004:** Results calculated for the parameters (preexponential factors, activation energies, and regression coefficients) governing the reactions of polysaccharides made up of hexoses and the derived products.

Reaction	Coefficient	ln k_0i_ (k_0i_, min^−1^)	Ea_i_ (kJ·mol^−1^)	R^2^
Gl_nf_ → GlPOS	k_Glnf_	29.0	116.4	<0.9
Gl_ns_ → GlPOS	k_Glns_	31.6	156.8	<0.9
GlPOS → Glu	k_GlPOS_	21.82	98.7	<0.9
Glu → HMF	k_Glu_	25.3	115.1	0.904
Glu → DP_Glu_	k_2Glu_	19.0	95.6	<0.9
Ga_n_ → GaPOS	k_Gan_	20.85	88.9	0.980
GaPOS → Gal	k_GaPOS_	30.62	127.9	0.949
Gal → HMF	k_Gal_	17.0	83.1	0.912
Gal → DP_Gal_	k_2Gal_	11.0	47.7	<0.9

* Susceptible glucan fraction: α_Gln_ = 0.05 g Gl_nf_/g Gl_n._

**Table 5 polymers-14-04732-t005:** Results calculated for parameters (preexponential factors, activation energies, and regression coefficients) governing the reactions of xylan and the derived products.

Reaction	Coefficient	ln k_0i_ (k_0i_, min^−1^)	Ea_i_ (kJ·mol^−1^)	R^2^
X_nf_ → XyPOS	k_Xnf_	62.89	266.2	0.999
X_ns_ → XyPOS	k_Xns_	44.32	180.7	0.999
XyPOS → Xyl	k_XyPOS_	47.13	195.9	0.987
Xyl → F	k_Xyl_	12.42	56.9	<0.9
Xyl → DP_xyl_	k_2Xyl_	10.58	43.2	0.992

* Susceptible xylan fraction: α_Xn_ = 0.85 g X_nf_/g X_n._

**Table 6 polymers-14-04732-t006:** Results calculated for the parameters governing the reactions of arabinosyl groups and the derived products.

Reaction	Coefficient	ln k_0i_ (k_0i_, min^−1^)	Ea_i_ (kJ·mol^−1^)	R^2^
Ar_n_ → ArPOS	k_Arn_	32.33	131.6	0.990
ArPOS → Ara	k_ArPOS_	8.23	37.3	0.969
Ara → F	k_Ara_	14.8	71.2	0.937
Ara → DP_Ara_	k_2Ara_	24.90	103.4	0.999

**Table 7 polymers-14-04732-t007:** Results calculated for the parameters governing the reactions of acetyl groups and acetic acid.

Reaction	Coefficient	Ln k_0i_ (k_0i_, min^−1^)	Ea_i_ (kJ·mol^−1^)	R^2^
Ac_n_ → AcPOS	k_Acn_	31.66	132.6	0.991
AcPOS → AcH	k_AcPOS_	8.52	39.9	<0.9

**Table 8 polymers-14-04732-t008:** Results calculated for the parameters governing the reactions of uronyl groups and the derived products.

Reaction	Coefficient	Ln k_0i_ (k_0i_, min^−1^)	Ea_i_ (kJ·mol^−1^)	R^2^
UA_n_ → UA	k_UAn_	3.09	25.6	0.943
UA → DP_UA_	k_UA_	24.85	102.4	0.903

## Data Availability

Not applicable.

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
