# Peer review of "Effects of Hydrothermal Processing on Miscanthus × giganteus Polysaccharides: A Kinetic Assessment"

_polymers, 2022, doi:10.3390/polym14214732_

Round 1
Reviewer 1 Report
The authors have described the detailed analyses and kinetics of hydrothermal processing of Miscanthus×giganteus for their biomass applications. These results will be helpful and informative for researchers in the field of polymer science and chemistry for biofuels. Whereas the reviewer thinks that the authors’ study in this manuscript is quite interesting, suggestive, and well-organized, some descriptions are not enough. The authors’ manuscript is not suitable for publication in “Polymers” in the present form. From these considerations, the reviewer recommends to accepting for publication in " Polymers," if the following issues are resolved.
1) What kind of “water” did the authors use in the autohydrolysis of the authors’ study? Although the authors have conducted a detailed assessment of chemical reactions, is not a grade of water used in the autohydrolysis process important in the authors’ study?
2) What kind of waste did the authors' process produce? How did the yields of the waste production depend on the processes? Did the existence of the waste influence autohydrolysis?
3) Were there any oligomeric saccharides after the autohydrolysis of the authors’ study?
4) Page 4, line 141, “Both types of samples were assayed for monosaccharides, furans, and organic acids by using the HPLC method described previously.”: The proper literature should be cited.
Author Response
PREVIOUS. The authors have described the detailed analyses and kinetics of hydrothermal processing of Miscanthus×giganteus for their biomass applications. These results will be helpful and informative for researchers in the field of polymer science and chemistry for biofuels. Whereas the reviewer thinks that the authors’ study in this manuscript is quite interesting, suggestive, and well-organized, some descriptions are not enough. The authors’ manuscript is not suitable for publication in “Polymers” in the present form. From these considerations, the reviewer recommends to accepting for publication in " Polymers," if the following issues are resolved.
QUERY 1) What kind of “water” did the authors use in the autohydrolysis of the authors’ study? Although the authors have conducted a detailed assessment of chemical reactions, is not a grade of water used in the autohydrolysis process important in the authors’ study?.
REPLY. We used tap water in our study. This information is given in the revised version of the article. Looking for a simpler and cheaper operational mode, we preferred tap water to deionized or distilled water. However, we would like to highlight that the origin of the water is not expected to have a significant influence on hydrothermolysis, in which the pH is the key variable. It can be noted that the slight acidity of tap water favors the hydrolytic breakdown of hemicelluloses, but this effect is of minor importance compared to the pH drop caused by the release of acetic acid generated from acetyl groups along the autohydrolysis reaction.
UERY 2) What kind of waste did the authors' process produce? How did the yields of the waste production depend on the processes? Did the existence of the waste influence autohydrolysis?
REPLY. Autohydrolysis is a fractionation method resulting in the production of a liquid phase (containing mainly saccharides of higher DP, monosaccharides, acetic acid, and sugar-decomposition products) and a solid phase (almost entirely made up of non-dissolved hemicelluloses, cellulose and lignin). The liquid phase is suitable (upon physicochemical refining for removing monosaccharides, acetic acid and sugar-decomposition products) to be used as prebiotic oligosaccharides. Alternatively, the liquid phase can be treated with acidic catalysts to yield chemicals such as furfural and derived compounds. On the other hand, the solid phase can be subjected to delignification treatments to yield soluble lignin fragments (suitable as antioxidants or as intermediates for polymer manufacture) and cellulose (which can be sold as such or employed as an intermediate for manufacturing multiple products), The literature indicates that the solids from autohydrolysis are more susceptible to delignification than the corresponding raw materials. On the basis of these ideas, it can be concluded that no spent fractions are produced in hydrothermal processing. The waste fractions or undesired byproducts would be obtained in the subsequent physicochemical or chemical treatments to be performed to obtain the target compounds.
QUERY 3) Were there any oligomeric saccharides after the autohydrolysis of the authors’ study?
REPLY. In the field of the hemicellulose/cellulose chemistry, the DP limits between oligosaccharides and polysaccharides are not clearly defined. Owing to this, we have included in a single group the higher saccharides (poly- and oligosaccharides, which include all the saccharides different to monosaccharides) derived from hemicelluloses and glucose. The DP distribution of the higher saccharides depends on the operational conditions (the higher severity, the lower DP distribution). The polymeric and oligomeric saccharides obtained in this study have been characterized in terms of the corresponding constituent units, and classified in the following groups: a) glucosyl units (GlPOS); b) xylosyl units (XyPOS), c) arabinosyl units (ArPOS), d) galactosyl units (GaPOS). This is the general method followed in literature to assess the breakdown of cellulose and hemicelluloses in autohydrolysis media.
QUERY 4) Page 4, line 141, “Both types of samples were assayed for monosaccharides, furans, and organic acids by using the HPLC method described previously.”: The proper literature should be cited.
REPLY. The information requested by Reviewer 1 has been included in the revised version of the manuscript (see text highlighted in red).

Reviewer 2 Report
Make the abstract relevant to the readers. I did not find any substantial details about the research work. Kindly, get it sorted. Avoid mathematical signs in the title. There is no point to dilate the reference list if it is not explained properly. Avoid the reference clusters ( page 2). The detail of raw material is insufficient. Hydrothermal processing without pressure measurement? How come? Kindly check it. Moreover, hydrothermal processing is an umbrella term, so what have you basically done in your work? Some details in the Result and discussion are irrelevant. Section 2.4 has no novelty. Kindly improve the modelling section.
Author Response
REPLIES TO REVIEWER2
QUERY 1) Make the abstract relevant to the readers. I did not find any substantial details about the research work. Kindly, get it sorted.
REPLY. The abstract has been modified following the reviewer´s intention (see text highlighted in red)
QUERY 2) Avoid mathematical signs in the title.
REPLY. The sign × included in the title (Miscanthus×giganteus) does not correspond to a mathematic sign, but to a standard biological nomenclature for hybrid specimens. Because of this, we would like to keep the title as it is, if possible.
QUERY 3) There is no point to dilate the reference list if it is not explained properly. Avoid the reference clusters (page 2).
REPLY. We intended to be exhaustive in citing all the available references dealing with autohydrolysis modeling. In fact, we tried to make a sound assessment (in fact, a “mini-literature review”) on all the studies reported in this subject. However, since different articles had different relevance for the purposes of this study, some of the references listed in the “clusters” are cited later in the manuscript, and others are not. For the sake of exhaustiveness, we would like to keep the text as it is. Not matter of this, we are open to cut a part of the references, if it is considered mandatory for acceptance.
QUERY 4) The detail of raw material is insufficient.
REPLY. In section 2.1, we indicated the origin of the raw material and the handling before processing and analysis. This is the type of information usually provided in articles dealing with biomass processing. If additional information is needed, we would need more details on how the text should be completed.
QUERY 5) Hydrothermal processing without pressure measurement? How come? Kindly check it.
REPLY. The total pressure of the system corresponds fairly with the one measured for pure water the at the same temperature (obtained from the liquid/vapour equilibrium data). In comparison with pure water, there are some factors increasing the total pressure of autohydrolysis reaction media (example: presence volatile components different from water, presence of residual air in the void volume of the reactor); but also other factors decreasing the total pressure are influential (example: presence of non-volatile compounds such as sugar oligomers and polymers, which behave according to the colligative properties of solutions). Both types of effects counteract each other, in a way that the total pressure observed for autohydrolysis media is close to the pressure corresponding to pure water at the considered temperature. This is why the total pressure is not cited in studies dealing with autohydrolysis.
QUERY 6) Moreover, hydrothermal processing is an umbrella term, so what have you basically done in your work?
REPLY. The experimental setup is simple: biomass is mixed with water and heated in a pressure-resistant reactor. The hot, compressed water is responsible for the effects observed. This is the usual mode in which the autohydrolysis reactions are carried out. This information is now given in the summary and in section 2.2 (see text highlighted in red).
QUERY 7) Some details in the Result and discussion are irrelevant.
REPLY. We have tried to be exhaustive in our explanations. We have revised our article, and we have not been able to identify irrelevant details. If correcting the Results and Discussion section by deleting some information is considered mandatory for acceptance, we would need additional information regarding the specific type of details to be omitted.
QUERY 8) Section 2.4 has no novelty.
REPLY. Section 2.4 summarizes one of the hypothesis employed in modeling (kinetic coefficients with Arrhenius-type dependence on temperature), and the procedure employed for solving the set of differential equations. As the reviewer says, the same hypothesis and calculation procedure have been employed in literature, but we believe that this information should be kept in the article in order to allow the readers to reproduce our results, if desired.
QUERY 9) Kindly improve the modelling section.
REPLY. Following the reviewer´s intention, and taking into account his/her previous query (Query 7), sections 3.3 and 3.4 have been modified to improve the readability of the text (see text highlighted in red).

Round 2
Reviewer 1 Report
As shown in the revised manuscript, some issues suggested by the reviewer were resolved.
The reviewer believes that the authors' findings will contribute to the advancement of polymer chemistry and polymer science.
The reviewer recommends accepting the revised manuscript for publication in “Polymers."
Reviewer 2 Report
I am not satisfied with the response provided for pressure measurement. Even the kinetic model also relies on the pressure component.